# Hyperactivation of MAPK Induces Tamoxifen Resistance in SPRED2-Deficient ERα-Positive Breast Cancer

**DOI:** 10.3390/cancers14040954

**Published:** 2022-02-14

**Authors:** Vasiliki Vafeiadou, Dina Hany, Didier Picard

**Affiliations:** 1Département de Biologie Moléculaire et Cellulaire, Université de Genève, Sciences III, 1211 Genève 4, Switzerland; vasiliki.vafeiadou@etu.unige.ch (V.V.); dina.hany@unige.ch (D.H.); 2On leave from: Department of Pharmacology and Therapeutics, Faculty of Pharmacy, Pharos University in Alexandria, Alexandria 21311, Egypt

**Keywords:** SPRED2, breast cancer, tamoxifen, endocrine resistance, ERα, ulixertinib, precision medicine

## Abstract

**Simple Summary:**

Tamoxifen has been used for more than 40 years to treat breast tumors that are dependent on the hormone estrogen for their growth. However, resistance and recurrence of the tumors during the course of the treatment are common. Understanding the mechanisms that drive tamoxifen resistance and discovering new biomarkers for early detection are keys for designing appropriate personalized therapies. Here, we show that low levels of SPRED2 may be useful as a novel biomarker of tamoxifen resistance. We found that SPRED2 deficiency causes a hyperactivation of the mitogen-activated protein kinases (MAPKs) ERK1/ERK2, which in turn enhances estrogen signaling and diminishes the toxic effects of tamoxifen on breast cancer cells. Treatment with the ERK1/2 inhibitor, ulixertinib, could restore their sensitivity to tamoxifen. Therefore, we propose that patients with estrogen-dependent breast cancer characterized by low expression levels of SPRED2 may be candidates for a combination therapy with tamoxifen and ulixertinib.

**Abstract:**

Breast cancer is the number one cause of cancer-related mortality in women worldwide. Most breast tumors depend on the expression of the estrogen receptor α (ERα) for their growth. For this reason, targeting ERα with antagonists such as tamoxifen is the therapy of choice for most patients. Although initially responsive to tamoxifen, about 40% of the patients will develop resistance and ultimately a recurrence of the disease. Thus, finding new biomarkers and therapeutic approaches to treatment-resistant tumors is of high significance. SPRED2, an inhibitor of the MAPK signal transduction pathway, has been found to be downregulated in various cancers. In the present study, we found that SPRED2 is downregulated in a large proportion of breast-cancer patients. Moreover, the knockdown of *SPRED2* significantly increases cell proliferation and leads to tamoxifen resistance of breast-cancer cells that are initially tamoxifen-sensitive. We found that resistance occurs through increased activation of the MAPKs ERK1/ERK2, which enhances the transcriptional activity of ERα. Treatment of SPRED2-deficient breast cancer cells with a combination of the ERK 1/2 inhibitor ulixertinib and 4-hydroxytamoxifen (4-OHT) can inhibit cell growth and proliferation and overcome the induced tamoxifen resistance. Taken together, these results indicate that *SPRED2* may also be a tumor suppressor for breast cancer and that it is a key regulator of cellular sensitivity to 4-OHT.

## 1. Introduction

Cancer is the primary cause of death worldwide, accounting for approximately 10 million deaths in 2020. Breast cancer (BC) represents the most common type of cancer, accounting for 2.26 million new cases in 2020 [1]. The fundamental mechanisms of BC development and progression have been extensively studied over the past years, yet BC patients are still facing high mortality rates [2]. Consequently, it is very important for new and potent therapeutic strategies to emerge.

BC is a highly polymorphic disease, driven by several molecular alterations. For that reason, there are various types of classifications that have been proposed to categorize breast tumors. Currently, in clinical practice, a substitute classification of the classical intrinsic classification [3] is being used, which characterizes the tumors based on histological characteristics and the expression of the hormone receptors estrogen receptor α (ERα) and progesterone receptor (PR), and of the human epidermal growth factor receptor 2 (HER2). According to this classification, there are five subtypes of BC, namely Luminal A (ERα+ and PR+; HER2–; Ki67 low), luminal B/HER2- (ERα+ and PR+; HER2-; Ki67 high), luminal B/HER2+ (ERα+ and PR+; HER2+), HER2-enriched (ER–, PR–, HER2+), and triple-negative BC (ER–, PR–, HER2–) [4].

The expression of ERα is of high significance in breast tumors, since it influences the prognosis of the disease and the appropriate therapeutic approach. Upon binding its cognate estrogenic ligand 17β-estradiol (E2), both genomic and non-genomic activities of ERα are triggered. For its genomic actions, ERα must dimerize and bind to specific DNA target sequences termed estrogen response elements (ERE) or to other transcription factors such as the activation protein-1 (AP-1) within the regulatory sequences of target genes. ERα dimers recruit various co-activator or co-repressor proteins that contribute to chromatin remodeling and transcriptional regulation. This affects the transcription of genes involved in BC development and progression, including genes that promote cell proliferation, migration and invasion, angiogenesis, and metastasis [5,6,7,8].

BC development and progression is highly dependent on the estrogen-ERα axis. For that reason, the suppression of estrogen synthesis with aromatase inhibitors, or the use of ERα antagonists, either selective ER modulators (SERMs) or selective ER downregulators (SERDs), have remained the pillars of ERα+ BC treatment for several decades [9]. Over the last 40 years, the SERM tamoxifen has been extensively and successfully used in clinical practice for endocrine therapy of ERα+ BC, which represent almost 70% of all BC cases [10,11]. The major active metabolite of tamoxifen, 4-hydroxytamoxifen (4-OHT), competes with estrogen for binding to the ligand binding domain of ERα and acts as a partial antagonist by inhibiting the C-terminal transcriptional activation function (AF2) associated with this domain [12,13]. Its binding alters the conformation of AF2, which promotes the recruitment of co-repressors rather than co-activators [14,15]. The problem is that a substantial proportion of BC patients, who are initially responsive to tamoxifen, ultimately acquire resistance with an accompanying recurrence of the disease [9]. Understanding the mechanisms of tamoxifen resistance is therefore of high significance for the design of new therapeutic strategies for successfully treating BC.

Although the molecular mechanisms underlying resistance to tamoxifen remain poorly understood, various mechanisms have been proposed [9,16], including the activation of the Ras/Raf/MAPK pathway [17]. The Ras/Raf/MAPK pathway regulates numerous cellular processes, including cell division, differentiation, and apoptosis [18,19]. The MAPKs ERK1/ERK2 can directly phosphorylate ERα, resulting in the stimulation of its estrogen-dependent transcriptional activity [20], or even in its transcriptional activation in the absence of its cognate ligand estrogen, the most extreme form of signaling crosstalk [18,21,22,23]. Ligand-independent activation of ERα was reported early on for several growth factors, which are known to trigger the MAPK cascade, including epidermal growth factor [21,24,25] and insulin [26,27]. The most prominent MAPK target site on ERα is serine 118 (S118), a site associated with the N-terminal activation function (AF1) of ERα [18,20,21]. Through the phosphorylation of S118, the impact of MAPK signaling on ERα activity both in the presence and in the absence of ligand may contribute to converting ERα-dependent BCs to hormone-independence and tamoxifen resistance [18,22,23].

The protein Sprouty Related EVH1 Domain Containing 2 (SPRED2) was first described 20 years ago as a negative regulator of the Ras/Raf/MAPK pathway [28]. It is widely expressed in various adult tissues, including the breast, and it can associate with Ras and neurofibromin to inhibit the activation of Raf and therefore to prevent the activation of MAPK and the phosphorylation of its downstream targets [29]. *SPRED2* is involved in the development and progression of various cancer types, and its expression has been shown to be positively correlated with a better outcome of the disease [30]. Indeed, its role as a tumor suppressor has been well established, as there is ample evidence that its expression is often downregulated in various cancer types [31]. However, its impact on BC has not yet been studied. In this study, we aimed to discover the role of SPRED2 in ERα+ BC cells and its impact on the outcome of endocrine therapy.

## 2. Materials and Methods

### 2.1. Cell Lines

The HEK293T cells were acquired from the American Type Culture Collection (ATCC). T47D cells of the European Collection of Authenticated Cell Cultures (ECACC) were purchased from Sigma-Aldrich. The MCF7-V cells were a gift from Dr. Wilbert Zwart (Netherlands Cancer Institute, Amsterdam, The Netherlands); their highly polymorphic short tandem repeat loci (STRs) were profiled using commercial services (ATCC and Microsynth) and found to be closely related (88%) to wild-type MCF7 cells; they were used here because they display more robust ERα responses than wild-type MCF7 cells. For comparison purposes, we used T47D cells, which have relatively low ERα protein levels and responses.

### 2.2. Cell Culture

The human breast carcinoma MCF7-V and the human embryonic kidney HEK293T cells were cultured in Dulbecco’s Modified Eagle’s Medium (DMEM) with phenol red (GIBCO, cat. # 10566016) and supplemented with 10% fetal bovine serum (FBS) (Sigma-Aldrich, cat. # F9665) and penicillin/streptomycin (Life Technologies, cat. # 15070063). Human T47D ductal carcinoma cells were cultured in RPMI-1640 with phenol red (GIBCO, cat. # 11875093), supplemented with 10% FBS, penicillin/streptomycin, sodium pyruvate (Life Technologies, cat. # 11360039), and 5 µg/mL recombinant human insulin (Sigma-Aldrich, cat. # I9278). For hormone deprivation, before each treatment, cells were cultured for 72 hours (h) in a medium without phenol red, complemented with 5% charcoal-treated FBS, penicillin/streptomycin and L-glutamine (Biowest, cat. # X0550). The cells were maintained in an incubator at 37 °C with 5% CO_2_ and were sub-cultured twice per week.

### 2.3. Cell Treatments and Proliferation Assays

Cells were treated with the vehicle ethanol (EtOH), 17β-estradiol (E2, Sigma-Aldrich, cat. # E2758), 4-hydroxytamoxifen (4-OHT, Sigma-Aldrich, cat. # H7904), fulvestrant (ICI 182,780, Sigma-Aldrich, cat. # I4409), or the ERK1/2 inhibitor ulixertinib (BVD-523, MedChemExpress, cat. # HY-15816). Note that ethanol (vehicle) always ended up being diluted ≥ 1:1000 in the tissue culture medium. For proliferation assays, cells were seeded on day 0 at a density of 25,000 cells per well into 24-well plates and harvested every 24 h for 5 days. For the 4-OHT dose–response curves, 1500 cells per well were seeded into 96-well plates and treated with 5 pM E2 and increasing doses of 4-OHT, for 14 days. For the ulixertinib dose–response curves with MCF7-V and T47D cells, 15,000 and 12,000 cells, respectively, were seeded per well into 96-well plates and treated with 5 pM E2 and increasing doses of ulixertinib for 72 h. For the pharmacological rescue experiments, 1500 cells per well were seeded into 96-well plates, treated with 5 pM E2 and increasing doses of 4-OHT in combination with 100 nM (for MCF7-V) or 300 nM (for T47D) ulixertinib for 14 days. For all other RT-qPCR, luciferase and immunoblotting experiments, the cells were treated with 100 pM E2, 100 nM 4-OHT and/or 100 nM ICI.

### 2.4. Lentiviral Transduction for sgRNA Gene Knockdowns and Generation of Single sgRNA Knockdown Clones

For lentiviral production, 4 × 10^6^ HEK293T cells were seeded in 10 cm dishes and co-transfected using the PEI MAX 40K transfection reagent (Polysciences Inc., cat. # 24765), with the plasmids pMD2.G, psPAX2 (gifts from Didier Trono’s laboratory at the EPFL, Lausanne, Switzerland), and a lentiviral vector for sgRNA expression. For gene knockouts, the sgRNA constructs were generated using the vector lentiCRISPRv2 (Addgene #52961) [32] and the following target sequences: for SPRED2 5′-AGTCTGAGGAGTCCACGTAG-3′, for the non-targeting negative control RPE65 5′-TTGTTAATGTCTACCCAGTG-3′. The selected sgRNA sequences correspond to the sequences used in the Human CRISPR knockout pooled library (Brunello) [33]. These sgRNAs were calculated to have the lowest off-target effects. Media-containing viral particles were collected at 24 h, 72 h and 96 h after transfection. Viral suspensions were filtered using a 0.45 μm filter and concentrated using 40% sterile polyethylene glycol. The viral suspensions were then rotated at 4 °C for 2 h, followed by centrifugation for 30 min at 4000× *g* at 4 °C to pellet the viral particles. After centrifugation, the supernatants were discarded, the pellets were resuspended in 750 μL complete culture medium (20 times concentrated relative to the initial viral suspension) and 200 μL was added to the MCF7-V cells in 10 cm dishes. After 24–48 h, cells were selected with 3 μg/mL puromycin (Cayman Chemical, cat. # 13884-50) for 4 days until the untransduced control cells had completely died. Single cells were selected by using a BD FACS Aria III Cell Sorter (BD Biosciences) and after expansion, SPRED2 protein levels were checked by immunoblotting. Finally, two clones were selected, for which the amount of SPRED2 was strongly reduced as compared to the parental MCF7-V. In parallel, one clone was selected as a negative control from the cells transduced with viral particles for RPE65 sgRNA, and where the SPRED2 protein level was similar to that of the parental cells.

### 2.5. Lentiviral Transduction for shRNA-Mediated Knockdowns

For the production of lentiviral particles, 4 × 10^6^ HEK293T cells were seeded in a 6 cm dish and co-transfected with the plasmids pMD2.G, packaging psPAX2, and the plasmids for expression of shRNAs. The latter were generated using the vector pLKO.1 (Open Biosystems) and the target sequences listed in Table 1. Media-containing viral particles were collected at 24 h, 36 h and 48 h after transfection. For viral concentration and cell transduction, the same procedure as before was followed. As negative controls, we used a scrambled sequence with no specific target (shScrambled) for MCF7-V and in HEK293T cells, and the empty pLKO.1 vector (shEV) for T47D cells.

### 2.6. Protein Extraction and Immunoblotting

To analyze the exogenously expressed ERα, HEK293T cells in 6-well plates were transfected transiently, using the PEI MAX 40K transfection reagent, with 1 μg of the expression vector for wild-type ERα (HEG0) [35] or for the ERα point mutant S118A (HE457) [36], and 1 μg of an expression vector for expression of enhanced green fluorescent protein (pEGFP-C1, Clontech) as a transfection control. Cells were harvested 42 h after transfection.

Cells were washed with Tris-buffered saline (TBS), detached with trypsin-EDTA (PAN Biotech, cat. # P10-024100), harvested in complete medium, and centrifuged at 1200× *g* rpm for 5 min. The pellets were washed twice with PBS and then lysed in an ice-cold lysis buffer (20 mM Tris-HCl pH 7.4, 2 mM EDTA, 150 mM NaCl, 1.2% sodium deoxycholate, 1.2% Triton-X-100), supplemented with a protease inhibitor cocktail (Thermo Fisher Scientific, cat. # A32953), and a phosphatase inhibitor cocktail (Thermo Fisher Scientific, cat. # 10668304) when studying phosphorylated proteins. Cell suspensions were sonicated for 15 min at high power with the Bioruptor sonicator (Diagenode). Cell lysates were centrifuged at 13,200× *g* rpm for 5 min at 4 °C. Supernatants were collected, cell debris were discarded, and protein concentrations were measured using the Bradford assay (Biorad, cat. # 5000006) at 595 nm. Then, 20–30 µg of proteins were used for the immunoblotting experiments. After gel electrophoresis and transfer to a nitrocellulose membrane, the membranes were blocked with 3% fat-free milk powder (or 3% bovine serum albumin for phosphoprotein detection) in TBS with 0.2% Tween-20 (TBS-T) for 1 h. Blocked membranes were incubated with specific primary antibodies in TBS-T overnight at 4 °C, washed 3× with TBS-T and incubated with the corresponding secondary antibody coupled to horse radish peroxidase (Agilent Dako) for 1 h at room temperature. After three washes of the membranes with TBS-T, protein bands were developed and visualized with an ECL kit (Enhanced ChemiLuminescence, Advansta, cat. # K-12045-D20) using an Amersham™ ImageQuant™ 800 biomolecular imager. The “Fisher BioReagents™ EZ-Run™ Prestained Rec Protein Ladder” from Thermo Fisher Scientific (cat. # BP3603-500) was used as protein molecular-weight marker for gel electrophoresis.

### 2.7. Antibodies

All antibodies that were used in this study are listed in Table 2 below.

### 2.8. Luciferase Reporter Assays

For this assay, MCF7-V, T47D, and HEK293T cells were seeded into 12-well plates. For estrogen signaling assays, cells were plated in phenol red-free media supplemented with 5% charcoal-stripped FBS. For ERE luciferase assays, cells were co-transfected with 1 μg EREtkLuc (XETL) [21] along with 50 ng of the Renilla luciferase expression plasmid pRL-CMV (Promega) as the transfection control, using the PEI MAX 40K transfection reagent. In addition, HEK293T cells were co-transfected with 1 μg HEG0 or HE457. To determine the transcriptional activity of a tumor-promoter regulatory element (TRE), MCF7-V cells were co-transfected with 1 μg of the luciferase reporter plasmid (TRE)_5_TL [37] along with 50 ng pRL-CMV. 24 h after transfection, cells were re-plated at a density of 10,000 cells per well in 96-well plates, and treated with 100 pM E2, 100 nM 4-OHT and 100 nM ICI as indicated. 24 h after treatment, cells were lysed and assayed for luciferase activities using the Dual Luciferase Reporter Assay kit (Promega, cat. # E1910). Firefly luciferase activities were standardized to the Renilla luciferase transfection control.

### 2.9. Wound Healing Assay

Cells were seeded at 90% confluency in 6-well plates. Cell layers were scratched using a 200 μL tip to form wounded gaps, washed with PBS and cultured in DMEM without phenol-red and complemented with 5% charcoal-treated FBS, penicillin/streptomycin and L-glutamine. Four different areas of the wounds were photographed at the time of the scratch (0 h) and after 24 h. Photographs were analyzed by measuring the open area of the wound using the software ImageJ [38].

### 2.10. RNA Extraction and Real-Time RT-qPCR

Total RNA was extracted from the cells using the guanidinium-acid-phenol TRI reagent containing 4M guanidium thiocyanate, 25 mM sodium citrate and 0.5% N-lauroylsarcosine, supplemented with 5% *v*/*v* β-mercaptoethanol [39]. The aqueous phase containing the total RNA was recovered after extraction with water-saturated phenol (Roth, cat. # A980.1) and RNA was precipitated with isopropanol. The RNA concentration was quantitated using a NanoDrop (Thermo Scientific). Using random primers (Promega, cat. # C1181) and GoScript reverse transcriptase (Promega, cat. # A5001), 400 ng of total RNA for each sample was reverse transcribed according to the instructions provided by the manufacturer. The synthesized cDNA was diluted in 2X GoTaq qPCR Master Mix (Promega, cat. # A6001) and combined with 1 μM each of the forward and reverse primers. The specific primer sequences used are listed in Table 3.

PCR reactions were run on a Bio-Rad CFX 96 Real-Time PCR instrument (Bio-Rad) with the following conditions: for the hot-start activation 2 min at 50 °C and 2 min at 95 °C, then for the subsequent cycles 10 s at 95 °C for denaturation and 40 s at 58 °C for annealing/elongation. Finally, the melting temperature of specific products was determined by sequentially increasing the temperature by 0.5 °C for 10 s from 65–95 °C. Relative gene expression was calculated by measuring the differences in the normalized Ct (ΔΔCt method), after normalization to the β-actin mRNA.

### 2.11. Crystal Violet Cell Proliferation Assay

For crystal-violet staining, cells were washed twice with PBS, fixed with a 4% formaldehyde solution in PBS, and stained with a 0.1% crystal violet solution in water for 30 min. After staining, the wells were thoroughly washed with water to remove excess background and left overnight to dry. The day after, crystals were dissolved in 100 µL glacial acetic acid and the absorbance was measured at 595 nm using a Sunrise plate reader (Tecan). Triplicate wells were assayed for each condition in at least three independent experiments.

### 2.12. In Silico Data Analyses

Percentages of *SPRED2* mutations in ERα+ breast cancer patients were obtained from the METABRIC database by using the cBioPortal web site [44,45,46]. Data for the methylation status of *SPRED2* in breast cancer patients were from the COSMIC database [47] (retrieved in April 2021). The combination index of ulixertinib with 4-OHT was calculated by the Chou-Talalay method [48] using the software CompuSyn [49].

### 2.13. Statistical Analyses

Unless otherwise indicated, the data shown are representative of three independent biological experiments with triplicate samples. Error bars indicate the standard errors of the means. Statistical significance was determined with a two-tailed unpaired t-test or a one- or two-way ANOVA, using the software GraphPad Prism (version 9) for Windows. *p*-values < 0.05 were considered statistically significant.

## 3. Results

### 3.1. Hypermethylation and Decreased Expression of SPRED2 Are Associated with Poor Clinical Outcome of ERα+ BC

To link *SPRED2* to BC, we began by examining the expression levels and the methylation status of *SPRED2* in BC patients. For this, we took advantage of the publicly available online databases COSMIC [47] and METABRIC [44,45,46]. The METABRIC database is one of the largest breast cancer cohorts to contain gene expression, mutation, and patient outcome data. An analysis of this BC dataset revealed that heterozygous deletions of the *SPRED2* gene occur in 7% of the ERα+ samples (103 from a total of 1459 samples) (Figure 1A). According to the COSMIC database, hypermethylation of the *SPRED2* gene occurs in approximately 40% of the BC patient samples (545 from a total of 1414 samples) (Figure 1B). The database indicates that the result of this analysis is largely based on methylation at position chr2:65313277, which lies in the 3′UTR and coincides with a CTCF binding site. Hypermethylation is commonly associated with decreased gene expression [50]. We therefore expect that *SPRED2* expression might be compromised in these samples. These data show that expression of *SPRED2* might be lower in a large portion of BC patients, suggesting that *SPRED2* might play a role as a tumor suppressor in BC as well.

To further assess the role and the prognostic value of *SPRED2* expression for the clinical outcome in BC patients, a survival analysis was conducted using the Kaplan–Meier plotter and the Gene expression-based Outcome for Breast cancer Online (GOBO) tool [51,52]. The algorithms, which these two tools use, sorted the samples according to the mRNA levels of a query gene (here *SPRED2*); they divide the samples into high and low expressors based on whether these levels are above or below the median of all samples. The overall survival (OS) (Figure 1C and Appendix A) and the relapse-free survival (RFS) (Figure 1D and Appendix A) were significantly enhanced in BC patients with high *SPRED2* mRNA levels. Low mRNA expression level of *SPRED2* also positively correlated with worse OS (Appendix A) and RFS (Figure 1E,F and Appendix A) in patients with ERα+ BC, and in those who have been treated with tamoxifen, suggesting that there could be a correlation between *SPRED2* expression levels and tamoxifen resistance. In ERα- BC, low *SPRED2* gene expression was positively correlated with better RFS (Figure 1G and Appendix A), although better OS was correlated with high *SPRED2* expression (Appendix A). Additionally. *SPRED2* expression positively correlated with distant metastasis-free survival (DMFS) in all BC patients, suggesting that *SPRED2* might affect the metastatic capacity of BC tumors regardless of ERα status (Appendix A). However, it is important to note, though, that for some of the Kaplan–Meier plots (Figure 1F,G and Appendix A), the *p*-values are higher than 0.05, possibly because of small sample sizes. Taken together, the methylation status and the expression levels of *SPRED2* could be clinically important parameters for BC progression and they could impact treatment outcome, especially in ERα+ BC patients.

### 3.2. SPRED2 Depletion Promotes the Proliferation of BC Cell Lines and Their Resistance to Tamoxifen

To explore the potential role of SPRED2, we depleted its levels in the ERα+ cell lines MCF7-V and T47D, and evaluated the cellular and molecular outcomes. We generated SPRED2 knockdown clones using lentiviral constructs to express short hairpin RNA (shRNA) and single-guide RNA (sgRNA) targeting *SPRED2* mRNA and gene, respectively (Figure 2A,B and Appendix A). Decreased SPRED2 expression in MCF7-V cells allowed the cells to proliferate faster than control cells, as measured by crystal violet staining assay (Figure 2C and Appendix A). The role of SPRED2 in cell migration was investigated using a wound-healing assay. The results showed that SPRED2 depletion significantly increased migration of MCF7-V cells compared to the control cells (Figure 2D,E and Appendix A). Therefore, our results support the role of SPRED2 as a tumor suppressor [31] and indicate that it is important in modulating cell proliferation and the migration of ERα+ BC cells.

Since the clinical data suggested that SPRED2 expression may be important for 4-OHT sensitivity (Figure 1F), we treated SPRED2-depleted cell lines (MCF7-V and T47D) with increasing concentrations of 4-OHT and evaluated cell proliferation with a crystal violet assay. In both cell lines, by comparison with the corresponding negative controls, the depletion of SPRED2 induced resistance to the cytotoxic effect of 4-OHT (Figure 2F–I and Appendix A). The extent of the acquired 4-OHT resistance was inversely correlated with the *SPRED2* mRNA levels (Figure 2A,F,G).

### 3.3. Knockdown of SPRED2 Stimulates ERα Activity by Activation of ERK1/2

In the context of MAPK signaling, it must be kept in mind that 4-OHT is a partial antagonist as it only blocks AF2 [12]. It can even be converted into a partial agonist, notably in cells in which the S118-associated AF1 is stimulated by growth-factor signaling see ref. [23] and references therein or upon cellular differentiation [53]. We hypothesized that the acquired resistance upon SPRED2 depletion could be due to signaling crosstalk with ERα. We noticed that SPRED2 depletion resulted in a major decrease in ERα protein levels in both MCF7-V and T47D cells (Figure 3A,B and Appendix A), as demonstrated by immunoblotting. This could be due to two possibilities, that either the ERα gene (*ESR1*) is less transcribed and thus less translated, or ERα is more active. Indeed, it is known that ERα, upon activation, becomes destabilized, leading to a more rapid proteasomal degradation [54]. This observation is indirect evidence that the depletion of SPRED2 might cause ERα to be more active and, thus, prone to degradation.

To investigate more directly whether the depletion of SPRED2 enhances ERα transcriptional activity, we performed luciferase reporter assays. We transfected MCF7-V and T47D cells with a reporter plasmid containing luciferase under the control of an ERE (EREtkluc) and measured the transcriptional activity of ERα under different conditions. This reporter assay revealed that the depletion of SPRED2 enhanced the E2-induced ERα activity (Figure 3C and Appendix A). The effect on the ligand-independent basal activity of ERα could not easily be seen under these experimental conditions, but reached a *p*-value of 0.058 in T47D cells (Appendix A; see also below for effects on endogenous target genes). The ligand-activated ERα activity was inhibited by 4-OHT and the pure antiestrogen ICI. Triggering the MAPK signal transduction pathway by adding insulin stimulated the ligand-independent basal activity of ERα, as well as its activities in the presence of E2 and 4-OHT. Therefore, we assume that the additional increase observed upon SPRED2 depletion may be due to the combinatorial effect of stimulating MAPK signaling with insulin and removing the MAPK signaling inhibitor SPRED2 (Figure 3C).

We next determined whether SPRED2 affects the expression of endogenous ERα target genes. By RT-qPCR, we found that the mRNAs of the ERα target genes *TFF1*, *CXCL12,* and *ADRB1*, were upregulated upon SPRED2 depletion; reduced levels of SPRED2 increased the basal levels, the E2-induced levels, and the levels in the presence of 4-OHT of the target gene mRNAs (Figure 3D–I and Appendix A). The latter is consistent with the notion that MAPK signaling can reveal the partial agonist function of 4-OHT more prominently [55]. The expression of the *ESR1* (ERα) gene itself was significantly reduced by SPRED2 depletion, implying that lower ERα protein levels are not entirely due to increased protein degradation resulting from increased transcriptional activity, but that SPRED2 depletion also affects the ERα mRNA levels (Appendix A). Conversely, we did not detect any impact of ERα activation or inhibition on the expression of *SPRED2* (Appendix A).

SPRED2 is known from the literature for its role in downregulating the activation of ERK [28]. For that reason, we thought that the Ras/Raf/MAPK pathway might be involved in the mechanism of 4-OHT resistance. We investigated the status of ERK1/2 in SPRED2-deficient cells by immunoblotting and found that ERK1/2 was highly phosphorylated in cells where SPRED2 was depleted (Figure 3A,B and Appendix A). The expression levels of the ERK target genes *ETV4* and *ETV5* [56] were significantly upregulated in these cells, as measured by quantitative RT-qPCR, confirming that ERK1/2 is more active in SPRED2-deficient cells (Appendix A). In addition, we performed a luciferase reporter assay to assess the activity of the transcription factor AP-1, a well-known ERK1/2 target [57]; this experiment revealed that AP-1, without any specific stimulation, is significantly more active after SPRED2 depletion (Appendix A). As a result, the increased activation of ERα could be due to its phosphorylation by ERK in a ligand-independent manner [21]. To test this, we co-transfected the negative control (shScrambled) and SPRED2-deficient HEK293T cells, which do not express ERα, with expression vectors for wild-type ERα or for the ERα S118A mutant, and with the EREtkluc reporter plasmid and an EGFP expression vector as the internal standard (Figure 3J,K). The protein levels of wild-type ERα were lower in the SPRED2-deficient cells, as shown by immunoblotting, indicating that ERα might be more active (Figure 3K,L). Remarkably, the protein levels of ERα S118A were not affected and were similar to the ones of the control (Figure 3K,L). Measuring the activity of ERα with a luciferase reporter assay revealed that SPRED2 depletion resulted in a significant additional stimulation of the E2-induced ERα activity, while this stimulation seems to be compromised for the S118A mutant (Figure 3M). However, we did not observe any increase in the basal ERα activity in SPRED2-depleted HEK293T cells, in contrast to what we had observed in BC cells. The latter could be due to different experimental conditions such as the cell line used or whether the ERα is expressed endogenously (BC) or by transient transfection (HEK293T). These results demonstrate that ERα is more active upon SPRED2 depletion and that the phosphorylation of S118 plays a critical role for this stimulation.

### 3.4. The ERK1/2 Inhibitor Ulixertinib Resensitizes SPRED2-Deficient BC Cells to 4-OHT

We carried out pharmacological rescue experiments to determine whether SPRED2-deficient BC cells can be resensitized to 4-OHT. The combination of endocrine therapy and ERK1/2 inhibitors has previously been suggested as a therapeutic approach to overcome endocrine resistance [58]. We used the ERK1/2 inhibitor ulixertinib to explore its effect on the tamoxifen resistance induced by SPRED2 deficiency. We treated cells with 4-OHT and ulixertinib either one alone or in combination. For each BC cell line, we treated the cells with low doses of ulixertinib so that cell proliferation would only be impaired by 10–20% (inhibitory concentrations IC10–IC20), as determined with pilot experiments, which incidentally revealed that T47D cells are intrinsically more resistant to ulixertinib (Appendix A). We treated the cells with ulixertinib in combination with increasing doses of 4-OHT to see if the tamoxifen-resistant cells would become sensitive again. Indeed, ulixertinib, at these relatively low doses, resensitized SPRED2-deficient BC cells to 4-OHT (Figure 4A,B and Appendix A). We calculated the synergy, with the Chou-Talalay method [48] using the software CompuSyn [49]. This revealed that the combination of ulixertinib with 4-OHT showed a very strong synergy (Combination index < 0.1), particularly in the SPRED2-depleted cells. Compared to the corresponding controls, the synergy was apparent across the whole range of the 4-OHT doses (Figure 4A,B). In addition to restoring 4-OHT sensitivity, ulixertinib was able to restore ERα protein levels to those of control cells (Figure 4C), as shown by immunoblotting. The increase in phosphorylation of ERK upon knocking down SPRED2 was also counteracted by ulixertinib (Figure 4C). Note that this somewhat counterintuitive effect of the ERK inhibitor on the phosphorylation of its direct target has been reported before [59]. Taken together, the results of our study suggest that a combination of ulixertinib and tamoxifen is worth exploring further.

## 4. Discussion

BC is the leading cause of cancer-related deaths worldwide. Endocrine therapy remains the foundation for the treatment of patients with ERα+ breast cancer [60,61]. Unfortunately, resistance to endocrine treatment occurs in a large proportion of patients with BC. About 40% of the women treated with tamoxifen for up to 5 years will present with recurrent BC within 15 years [10]. Understanding the mechanisms of resistance initiation and progression and the identification of new therapeutic targets or more specific biomarkers that could predict the outcome of therapy are critical for the successful treatment of resistant tumors.

The Ras/Raf/MAPK signaling pathway is abnormally activated in various tumors to support the survival and proliferation, and thus, targeting this pathway may be beneficial [62,63,64,65]. SPRED2, a negative regulator of the MAPK signal-transduction pathway, is frequently downregulated in cancers such as hepatocellular carcinoma and prostate cancer [66]. SPRED2 protein levels are strongly inversely associated with tumor progression and metastasis. In colorectal cancer and hepatocellular carcinoma, SPRED2 can inhibit the epithelial-to-mesenchymal transition and cell motility [67,68]. In our study, we found that *SPRED2* can act as a tumor suppressor in BC as well, and that decreased levels of SPRED2 can increase their cell proliferation and migration capacity.

The Ras/Raf/MAPK pathway is fundamental for controlling major cellular processes, such as cell survival and proliferation. Upon induction by a variety of extracellular stimuli, ERK1/2 becomes phosphorylated at threonine and tyrosine residues, resulting in its nuclear transfer and activation. This signal transduction pathway mediates growth responses by activating key transcription factors such as c-Jun and c-Fos of the AP-1 family [69]. These transcription factors bind to AP-1 binding sites, known as 12-O-tetradecanoylphorbol-13-acetate (TPA) response elements (TRE), to induce the transcription of genes implicated in cell cycle progression and other cellular processes [70]. Activated ERK1/2 can also phosphorylate S118, located in the AF1 domain of ERα [21]. This can trigger the ligand-independent activation of ERα as a transcription factor or can further stimulate its E2-stimulated transcriptional activity. In either case, ERα may also regulate target genes by tethering to TRE-bound AP-1 [71,72].

These events can affect the response to endocrine therapy. Abnormally activated ERK can increase ERα sensitivity to estrogen and thereby contribute to tamoxifen resistance [73,74,75]. In addition, AP-1 holds a central role in BC cell proliferation and the crosstalk between ERα and AP-1 was found to be a pivotal factor in BC progression [71,72]. Clinically, ERα+ BC tumors that exhibit tamoxifen resistance and increased invasiveness have been associated with upregulated AP-1 activity [76,77,78]. Thus, tamoxifen resistance can emerge from increased levels of activated ERK1/2, AP-1 activity, phosphorylated ERα, and the increased transcription of ERα target genes [79]. In this study, we found that cells expressing lower amounts of SPRED2 exhibit increased levels of phosphorylated ERK1/2, which lead to hyperactive ERα and AP-1, as shown by luciferase reporter assays, and higher transcription of several ERα target genes. We found that in SPRED2-deficient BC cells, ERα is also activated in a ligand-independent manner, most likely by the direct ERK1/2 phosphorylation of ERα on S118 since changing that site to alanine (S118A) abolished the response. As a result, cells may acquire an increased proliferation capacity and develop tamoxifen resistance.

In the past, *SPRED2* has been linked to drug resistance [80] and described as a potential prognostic marker for multiple cancer types such as chronic myeloid leukemia [80,81], prostate cancer [66], and hepatocellular carcinoma [82]. According to our findings, *SPRED2* is a potential biomarker for endocrine therapy resistance in ERα+ BC patients.

Several studies on the role of the ERK pathway in tumor progression have used MEK inhibitors, such as PD98059 or U0126 [83,84]. Here, we have used ulixertinib, a novel ERK1/2 kinase inhibitor, which has been successfully used to treat several models that exhibit intrinsic or acquired resistance to other ERK pathway inhibitors [85]. Ulixertinib is currently the only ERK1/2 inhibitor that is in clinical trials (phase 1/2) for advanced malignancies [86,87]. In our experiments, the treatment of SPRED2-deficient BC cells with ulixertinib restored normal ERα protein levels as well as normal (lower) levels of ERK1/2 phosphorylation, and resensitized the cells to tamoxifen. Based on the results of our study, it is conceivable that patients with lower expression levels of *SPRED2* could be candidates for combination therapy consisting of tamoxifen and ulixertinib. Our mechanistic studies indicate that lower SPRED2 protein levels are a relevant parameter, unless *SPRED2* mRNA levels can be robustly used as a proxy. Rigorous preclinical studies will be needed to develop clear criteria for “low levels” and to determine whether our results can be translated to a clinical setting, and whether this combination therapy can ultimately become a novel therapeutic choice for ERα+ BC patients with reduced *SPRED2* expression levels.

Exploring and comprehending the molecular mechanisms of certain tumors and how they acquire resistance to specific treatments is essential to apply precision medicine. Each patient’s cancer carries different types of mutations, which can have an impact on the prognosis of the disease, on the outcome of the selected treatment and on the development of resistance. The combination of several FDA-approved molecules for a better response to therapy is now becoming the preferred type of treatment in oncology. Personalizing the treatment may help to solve issues of cancer recurrence and resistance to treatment.

## 5. Conclusions

In summary, what we could conclude from our study is that *SPRED2* deficiency can induce resistance to tamoxifen in ERα+ BC cells. Its depletion leads to increased ERK phosphorylation and activity, and stimulates ERα, which can stimulate cell proliferation and contribute to tamoxifen resistance. The combination of 4-OHT with the ERK inhibitor ulixertinib might eventually be a successful therapeutic choice for SPRED2-deficient ERα+ BC (Figure 4D). Our results highlight *SPRED2* as a potential prognostic marker of tamoxifen resistance in ERα+ BC patients, contribute to the existing knowledge about the mechanisms of drug resistance, and offer additional avenues for new personalized therapies.

## Figures and Tables

**Figure 1 cancers-14-00954-f001:**
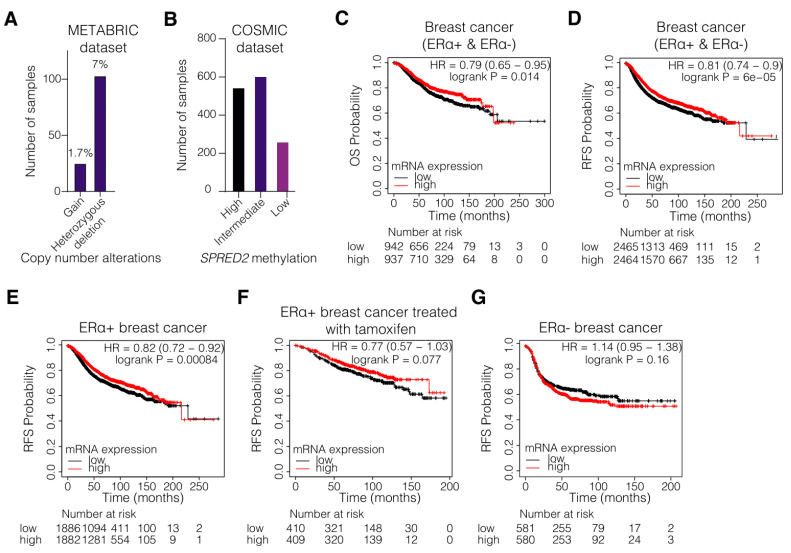
Low SPRED2 gene expression correlates with poor survival of patients with breast tumors. (**A**) METABRIC dataset analysis showing the mutation types of SPRED2 in ERα+ breast cancer patients. (**B**) COSMIC dataset analysis showing the SPRED2 methylation status in breast cancer patients. Of the total number of 1414 samples in this dataset, all but one of the 545 samples are hypermethylated at position chr2:65313277 (located in the 3′UTR and coinciding with a CTCF binding site). (**C**–**G**) Kaplan–Meier plots for the overall survival (OS) and the relapse free survival (RFS) of all (C,D), ERα+ (**E**), tamoxifen-treated ERα+ (**F**), and ERα– (**G**) breast cancer patients, classified as tumors expressing high levels (red line) and low levels (black line) of SPRED2 mRNA.

**Figure 2 cancers-14-00954-f002:**
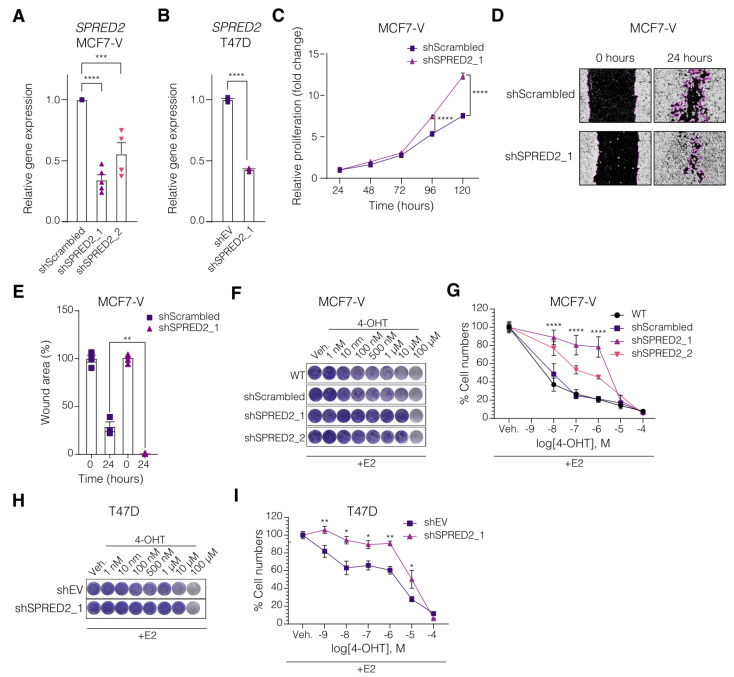
SPRED2 deficiency enhances cell proliferation, migration, and induces resistance to 4-OHT in ERα+ BC cells. (**A**,**B**) Bar graphs showing the RT-qPCR quantitation of SPRED2 mRNA levels in cells transduced with viral particles for the expression of two different shRNAs targeting SPRED2 or of a negative control shRNA in MCF7-V (shScrambled) (**A**) and of the empty vector in T47D (shEV) (**B**) cells. (**C**) Relative proliferation of MCF7-V cells expressing shScrambled and shSPRED2, measured with a crystal violet assay in monolayer culture. The numbers of cells for each of them are standardized to the corresponding values at 24 h set to 1. (**D**) Cell migration assay with MCF7-V cells with and without SPRED2 knockdown. Representative images of the scratch wound areas, outlined with magenta, at the time of the wounding (0 h) and after 24 h. Images of all four replicates (including these) are presented in Appendix A. (**E**) Bar graphs showing the means of closure for MCF7-V cells. The extent of closure was analyzed with the software ImageJ. (**F**) Cell growth analyzed by crystal violet staining of MCF7-V cells with and without SPRED2 knockdown and treated with increasing doses of 4-OHT. (**G**) Dose–response curves of MCF7-V cells treated with increasing doses of 4-OHT. WT, parent MCF7-V cells. (**H**) Cell growth analyzed by crystal violet staining of T47D cells treated with increasing doses of 4-OHT. (**I**) Dose–response curves of T47D cells treated with increasing doses of 4-OHT; for panels (**F**–**I**), *n* = 3 independent experiments in triplicates. All error bars represent the standard errors of the means (mean ± SEM). Asterisks indicate significant differences compared to the samples with shScrambled (MCF7-V) or shEV (T47D) (ns for *p* > 0.05, * *p* ≤ 0.05, ** *p* ≤ 0.01, *** *p* ≤ 0.001 and **** *p* ≤ 0.0001). Statistical significance was determined with a one-way ANOVA (**A**), a two-tailed unpaired *t*-test (**B**,**E**), and a two-way ANOVA with Bonferroni multiple comparison test for all other panels.

**Figure 3 cancers-14-00954-f003:**
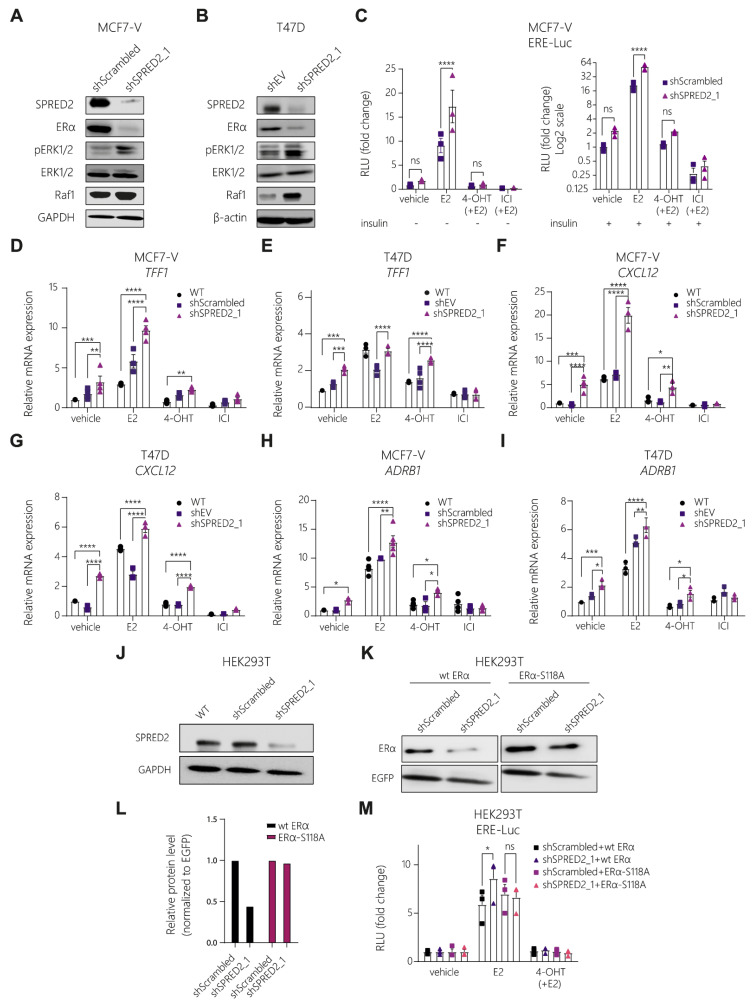
Knockdown of SPRED2 stimulates ERα transcriptional activity by increasing the activity of the MAPKs ERK1/2. (**A**,**B**) Immunoblots for the indicated proteins of MCF7-V (**A**) and T47D (**B**) SPRED2 knockdown and control cells. (**C**) Luciferase reporter assays for endogenous ERα activity with MCF7-V cells transiently transfected with the ERE-Luc reporter plasmid. The relative luciferase activities (RLU) are relative to the activities of the internal transfection standard Renilla luciferase and the vehicle-treated shScrambled control samples set to 1. Graphs are based on *n* = 3 independent experiments. (**D**–**I**) Expression of ERα target genes in MCF7-V and T47D cells; mRNA levels were analyzed by RT-qPCR following 20 h (for TFF1) and 6 h (for CXCL12 and ADRB1) of treatments as indicated. WT, parent MCF7-V cells. The data were normalized to the values of the corresponding vehicle-treated untransfected cells set to 1; *n* ≥ 3 independent experiments. (**J**) Immunoblot showing the knockdown of SPRED2 in HEK293T cells with GAPDH as internal standard. WT, non-perturbed HEK293T cells. (**K**) Immunoblot showing wild-type and S118A mutant ERα protein levels in HEK293T cells after transient transfection with expression vectors HEG0 and HE475, respectively, using EGFP as a transfection control. (**L**) Quantification of the immunoblot shown in panel (**K**). Protein levels are normalized to the levels of co-expressed EGFP and shown relative to those of shScrambled-expressing cells. Protein levels were analyzed with ImageJ. (**M**) Luciferase reporter assays for exogenous ERα activity with HEK293T cells; wild-type (wt) ERα and the ERα point mutant S118A were expressed from plasmids HEG0 and HE457, respectively. The relative luciferase activities (RLU) are relative to the activities of the internal transfection standard Renilla luciferase and the vehicle-treated samples set to 1. Graphs are based on *n* = 3 independent experiments. Immunoblot images of panels (**A**,**B**,**J**,**K**) are representative images. In bar graphs, all error bars represent the standard errors of the means (mean ± SEM). Asterisks indicate significant differences (ns for *p* > 0.05, * *p* ≤ 0.05, ** *p* ≤ 0.01, *** *p* ≤ 0.001 and **** *p* ≤ 0.0001). Statistical significance was determined with a two-tailed unpaired t-test (**C**) and a two-way ANOVA with Bonferroni multiple comparison test for all other panels. Uncropped images of immunoblots with molecular weight standards are in Appendix A.

**Figure 4 cancers-14-00954-f004:**
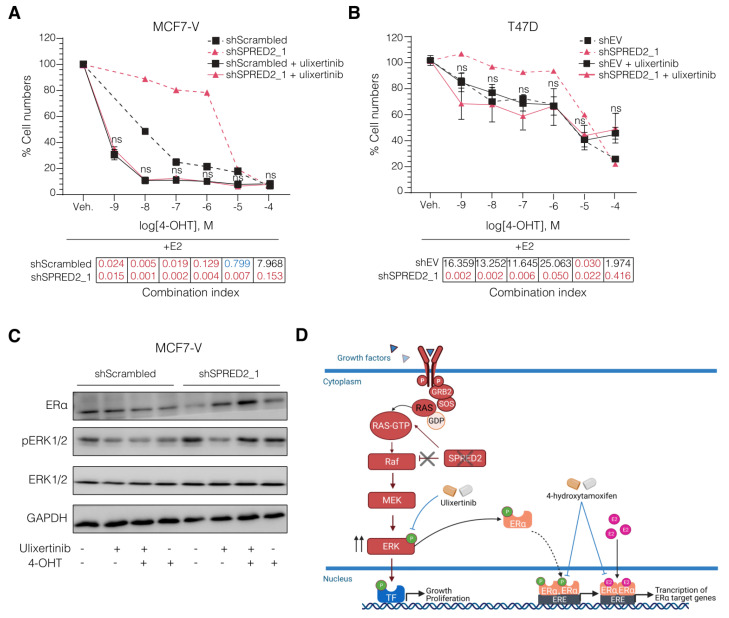
Ulixertinib resensitizes SPRED2-deficient BC cells to 4-OHT. (**A,B**) Dose–response curves with increasing doses of 4-OHT in combination with ulixertinib at 100 nM for MCF7-V (**A**) and 300 nM for T47D (**B**). The dashed curves indicate the response to 4-OHT without ulixertinib, copied from Figure 2G,I for comparison. Statistical significance was determined with a two-way ANOVA with Bonferroni multiple comparison test for both panels. All error bars represent the standard errors of the means (mean ± SEM). Significance is indicated compared to the shScrambled (MCF7-V) or shEV (T47D) samples, each treated with ulixertinib (ns for *p* > 0.05). Tables below the curves indicate the combination index of 4-OHT with ulixertinib for each dose, categorized as very strong synergism (red), moderate synergism (blue), and strong antagonism (black), calculated by the Chou-Talalay method with the software CompuSyn. (**C**) Representative immunoblots showing the changes in ERα and p-ERK protein levels of MCF7-V cells after treatment with 4-OHT and ulixertinib alone or in combination. (**D**) Schematic representation of the proposed mechanism driving 4-OHT resistance in SPRED2 deficiency and how to overcome it. The illustration was created with BioRender.com. Uncropped images of immunoblots with molecular weight standards are available in Appendix A.

**Table 1 cancers-14-00954-t001:** shRNA target sequences.

shRNA Name	Target Sequence 5′–3′	TRC Clone ID ^2^
shSPRED2_1	CAACAGCTACAGACAGTTCTT	TRCN0000056832
shSPRED2_2	GCAATCGAAGACCTTATAGAA	TRCN0000056828
shScrambled	CCTAAGGTTAAGTCGCCCTCG ^1^	

^1^ Sequence is from a previous publication [34]. ^2^ TRCN, the RNA consortium number (https://www.broadinstitute.org/rnai/trc accessed on 19 May 2020).

**Table 2 cancers-14-00954-t002:** Antibodies used in this study.

Antibody	Supplier	Working Dilutions
Anti-ERα	Bethyl Laboratories (cat. # A300-498A)	1:1000
Anti-SPRED2	Sigma-Aldrich (cat. # S7320)	1:1000
Anti-GAPDH (6C5)	Abcam (cat. # ab8245)	1:30,000
Anti-ERK 2 (C-14)	Santa Cruz Biotechnology (cat. # sc-154)	1:200
Anti-p-ERK (E-4)	Santa Cruz Biotechnology (cat. # sc-733)	1:1000
Anti-RAF1 (C-12)	Santa Cruz Biotechnology (cat. # sc-133)	1:500
Anti-β-actin	Millipore (cat. # MABT825)	1:5000
Anti-EGFP	Roche (cat. # 11814460001)	1:5000
IgGs, rabbit and mouse	Sigma-Aldrich (cat. # I5006 and I5381)	1: 10,000–1: 30,000

**Table 3 cancers-14-00954-t003:** Primer sequences used for RT-qPCR experiments.

Gene	Forward Primer 5′–3′	Reverse Primer 5′–3′
*SPRED2*	GGGACAGGCGTCTAGGTGAAC	AAAGCCGCTTCGTCCATTGC
*CXCL12*	CCCAGGTGCTACACCCTTTT	CAGGAATGGGGCTCCTTCAG
*ADRB1*	CCGGGAACAGGAACACAC	GAAAGCAAAAGGAAATATGTC
*TFF1* ^1^	CAATTCTGTCTTTCACGGGG	CACCATGGAGAACAAGGTGA
*ESR1* ^2^	GCTCTTGGACAGGAACCAGG	AAGATCTCCACCATGCCCTCT
*β-actin* (normalization control) ^3^	CATGTACGTTGCTATCCAGGC	CTCCTTAATGTCACGCACGAT
*ETV4* ^4^	AGGAACAGACGGACTTCGCCTA	CTGGGAATGGTCGCAGAGGTTT
*ETV5*	GGGTTCTTTGGGGTTTGTTT	CGCAGGGGAAAGTATTTCAA

These sequences are from previous publications: ^1^ [40], ^2^ [41], ^3^ [42], ^4^ [43].

## Data Availability

The datasets used and/or analyzed during the current study are available from the corresponding author on reasonable request.

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
