# Peer review of "Hyperactivation of MAPK Induces Tamoxifen Resistance in SPRED2-Deficient ERα-Positive Breast Cancer"

_cancers, 2022, doi:10.3390/cancers14040954_

Round 1

Reviewer 1 Report

The manuscript deals with the issue of MAPK influence on Tamoxifen action in Breast Cancer ER alpha positive and SPRED 2-deficient.

The influence of RAS-RAF-MAPK pathways on tamoxifen and chemotherapy has been already approached by other group, deregulated MAPK pathways helps to develop resistance to tamoxifen. In this manuscript authors, searching for new biomarkers and therapeutic approaches, found that an inhibitor of MAPK signal, SPRED2, is down regulated in a large proportion of breast cancers patients. Authors found that the SPRED2 involvement in tamoxifen resistance is coherent with its role in regulating ERK signalling.  Indeed, authors demonstrate that SPRED2 deficiency causes iperactivation of ERK1/2 and increases estrogenic signalling while decreases the toxic effect of tamoxifen causing resistance with failure of treatment.

The data presented in this paper are quite interesting from the point of view of mechanisms as well as of novel therapeutic strategies. The manuscript must be reconsidered by authors with greater care for the reasons listed below.

General and important remark

Supplementary Figures are presented by authors not as Figures which closely report the data but are western blots representing more than one results and often it is hard to find between them the results cited in the text. Almost every result of many figures in the text is supported by supplementary data that is hard to find as the supplementary file. This weakens the main data presented in the paper.

Please compose the supplementary figures as expected by readers and most important as stated by authors in the text.

This applies for all supplementary figures cited in the text that are unreadable and invaluable in the present form. Some data of the supplementary figures were also impossible to find.

Specific points to be raised:

  • Fig2C, legend title is “Relative proliferation of MCF7-V cells expressing shScramble and shSPRED2 ……..”. The values are expressed as fold change respect to what? Wild type un-transfected MCF7-V is the control? Please clarify in the legend.
  • Fig2D on the left. Authors are invited to comment why the shSPRED2- depleted cells take 3 days to express differences in proliferation respect to shScrambled cells.
  • Fig2D on the right. How the scratch wound areas of MCF7-V control untreated cells is? The quantification of this sample gives an idea on how infection per se changes the cell migration properties.
  • Fig3A, B The use of proteasomic inhibitors such as MG132 or Lactacystin could have help to evaluate the proteasomic component in the mechanism of ERalpha decrease induced by silencing of SPRED2. In C, T47D cells depleted of SPRED 2 are compared with shEV instead to shScrambled as MCF7-V. There is a particular reason to avoid comparison with shScrambled cells.

In A and B the protein bands of these WB lack of standard reference that, although presented in the uncropped WB of Supplementary file,  it is commonly accepted to repeat the standard reference in the cropped WB of main text  figures.

Quantifications of blots should be presented as means+/-STD of the replicate performed, with associated statistical analyses. This is stated in the instruction for authors.

Fig 3C on the Right. The results of this experiment is not clear and not coherent with the conclusion in the text: “The effect on the ligand-independent basal activity of ER alpha could not easily be seen under these experimental conditions, but reached a p-value of  0.058 in T47D cells (Figure S3A)”  . Figure S3A does not exist as Figure between supplementary data. Authors are invited to present the supplementary Figure S3A and to ameliorate this figure for better understanding of readers.

  • Lines 389-391 please better clarify this sentence :” we assume that the additional increase observed upon SPRED2 depletion may be  due to the combinatorial effect of stimulating MAPK signalling with insulin and removing  the MAPK signalling inhibitor SPRED2 (Figure 3C).
  • Fig S4AC and S4D. Honestly, it is impossible to find and evaluate these figures.
  • Fig3 J and K Reference standard for each band of the blots and quantifications of blots should be presented as means+/-STD of the replicate performed, with associated statistical analyses. See point 3.
  • Fig4A the SPRED2-deplited cells are compared with ulixertinib treated SPRED2-deplited cells as well as shScrambled cells with those ulixertinib- treated. In the figure B T47D cells show that ulixertinib does not change the response to Tamoxifen of Scrambled cells while does it in the SPRED2 depleted cells. In the MCF7-V the ulixertinib has effect in the SPRED2 depleted cell line as well in the Scrambled cells although with appreciable differences between the two cellular systems. As matter of fact, in MCF7_V the ulixertinib in shScrambled cells has effect on tamoxifen treatment although this response is more evident with SPRED2 silencing. In these cells ulixertinib seems to have some effect independent on SPRED2. If this is the case authors should comment this.
  • Fig 4 C Reference standards for each band of the blots and quantifications of blots should be presented as means+/-STD of the replicate performed, with associated statistical analyses, see point 3. In addition authors in the text say: “….ulixertinib was able to restore ERα protein levels to those of control cells (Figure 4C), as shown by immunoblotting” . What and where control is in these experiments? ShScrambled and SH SPRED2_1 are the only samples shown. There does not appear any untreated MCF7-V sample. These results is part of the main conclusion of this paper and must be clearly presented.
  • The large part of discussion deal with topics useful to introduction. The paper is full of data which deserve to be well commented and not only reported.
  • Supplementary Materials: The following are available online at https://www.mdpi.com/..??. Figure S1: Low SPRED2 gene expression correlates with poor survival of patients with breast tumors; Figure S2: SPRED2 deficiency enhances cell proliferation, migration, and induces resistance to 4- OHT in ERα+ BC cells; Figure S3: Knockdown of SPRED2 stimulates ERα transcriptional activity by  increasing the activity of the MAPKs ERK1/2; Figure S4: Ulixertinib resensitizes SPRED2-deficient  BC cells to 4-OHT.

It was hard to find  S1-S4 figures in the file containing this material which looks more properly uncropped Western Blot.

Reviewer 2 Report

I read with interest this article which highlights the interest of SPRED2 in RE+ breast cancer.

Just a few remarks

Line 52-55 : in clinical practice, Ki67 is also used for this classification

Line 302 – 319 : the Kaplan Meyer curves do show a potential role for SPRED2 in survival, but are the other factors influencing survival of breast cancer patients taken into account? A multivariate analysis would be required to formally conclude. This should be specified at least in the discussion.

I think you cannot conclude that SPRED2 could be a prognostic marker since you did not study its expression according to other prognostic factors in breast cancer. For that it would have been useful to use other cell lines known to be more "aggressive" with higher migration potentials for example MDA-MB-231 or 435s and simply quantify SPRED2 but the best is to study the expression on human cancer tissue.

Reviewer 3 Report

The manuscript by Vafeiadou et al. documents the function of SPREAD2 in the context of breast cancer. Using genetic approaches, cell line models and small molecule inhibitors, they convincingly show that SPREAD2 modifies Estrogen Receptor responses through the regulation of the ERK MAP-kinase pathway. In addition they document the role of SPREAD2 in Tamoxifen resistance, and the correlation of SPREAD2 expression and survival of BC patient cohorts.

The manuscript is well written, the experiments are adequate regarding the questions raised and the results convincing. Since the study relates to oncogenic signaling and to resistance to treatment, it is of interest to a wide audience.

Questions to address:

  1. Figure 1B shows quantification of methylation at SPREAD2 locus. The authors should indicate more details on how the analysis was done: did it include the gene including intron sequences, the promoter and distant regulatory elements?
  2. Figure 1, the threshold used to define low and high mRNA expression tumors should be specified in the results section and/or in the figure legend.
  3. Figure 2D and E show the results of wound healing assay on MCF7 cell line. Do T47D cells, used in all other assays in addition to MCF7, support the results? Since the functional assays are done on cell lines, it is essential to know whether the results obtained are dependent on a specific cell line or shared by several.
  4. Figure 3, panel M is not described in the figure legend.
  5. Lanes 474-476, the sentence “the results of our study suggest that a 473 combination of ulixertinib and tamoxifen may be a novel therapeutic choice for ERα+ BC 474 patients with reduced SPRED2 expression levels.” should be revised as the data may support an interest for a combination therapy, but at this stage, it remains in vitro observations that require preclinical studies before concluding with such statement.
  6. Why is SEM used throughout the figures instead of SD -which, unlike SEM, has the advantage of illustrating the variability of the observations- ?

Reviewer 4 Report

The manuscript titled “Hyperactivation of MAPK induces tamoxifen resistance in SPRED2-deficient ERα-positive breast cancer” is describes the authors had used online databases to link SPRED2 to breast cancer, and then the authors noticed about 40% of the breast cancer patients with hypermethylation of SPRED2 gene might result in lower expression of SPRED2 in the portion of breast cancer patients. And the overall survival and relapse-free survival were significantly enhanced in breast cancer patients with high SPRED2 mRNA levels. The authors further use two breast cancer cell lines MCF7-V and T47D to investigate the mechanism of relatively higher SPRED2 levels leading to good outcomes of breast cancer patients by inhibiting the MARK signaling pathway to inhibit cancer cells proliferation and overcome the induced tamoxifen resistance of ERα-positive breast cancer.

Overall, the manuscript is well-written for the most part. As the authors stated even though the impact of SPRED2 on breast cancer has not yet been studied, the role of SPRED2 as a negative regulator of the Ras/Raf/MAPK pathway and the expression of SPRED2 leads to better outcomes of various cancer types has been well documented, breast cancer is just one of those cancer types. Based on the online database, the authors realized that the high SPRED2 mRNA levels in breast cancer patients resulted in better outcomes. So the novelty is the top concern of this study. Despite the novelty, the followings are some concerns and the comments have been pointed out that the authors may want to consider.

Major Concerns and Comments:

1:        Line 112-116: The authors indicated MCF7-V cells display more robust ERα responses than wild-type MCF7 cells. Please provide the related references or the related data you have done. Have you ever done the experiments with wild-type MCF-7 or used other breast cancer cell lines as low ERα responses control. The wild-type MCF-7 data should give more credits for your study.

2:        Line 257: The authors run Real-Time PCR with 49 cycles which is too much than the normal protocol. Would you please provide some more information on why you chose 49 cycles?

3:        Line 305-307: The authors described that relapse-free survival was significantly enhanced in BC patients with high SPRED2 mRNA levels. How about the protein level?

4:        Line 430-431: What do you mean a two-way ANOVA was performed for all the panels including Figure 3C as you mentioned was determined with a two-tailed unpaired t-test? Please provide some more detailed information for the statistics.

5:        Page 11 of Figure 3: Would you please provide some details why some groups seem only with 1 sample, some with 2 samples, some with 3, 4, or up to 5 samples?

6:        From this study, the authors would like to point out a novel biomarker as early prognostic in ERα-positive breast cancer patients. And fortunately, the authors noticed that SPRED2 is downregulated in a large proportion of breast cancer patients from the online database. The patients with low expression levels of SPRED2 may be the candidates for combination therapy with tamoxifen and ERK1/2 inhibitor ulixertinib. The question is how to define the SPRED2 expression is at a low level? Protein level or mRNA level?

7:        Based on the authors’ hypothesis, the hyperactivation of MAPK (ERK1/ERK2) induces tamoxifen resistance of ERα-positive breast cancer patients due to the lower level of SPRED2. Do the authors have any idea to use the hyperactivated MARK as a biomarker instead of SPRED2?

Minor Concerns and Comments:

1:         Line 131: The vehicle ethanol concentration should be mentioned for the cell treatment.

2:         Line 283, line 316, line362, line 429-430, and line 485, etc. all through the manuscript include the supplementary figure legends: I’d suggest using the p italic as it refers to a p-value.

3:         Figure S2A: Please provide the reason why you chose GAPDH as an internal control for MCF-7 testing and β-actin was used as a control for T47D experiments to detect the same protein SPRED2. Both MCF-7 and T47D cancer cell lines express GAPDH and β-actin. Why you did not use the same internal control?

4:         Figure 4C: Protein marker labels are missing.

5:         I’d suggest the authors provide necessary figure legends for all your original images.

6:         Line 429: The authors defined untransfected MCF7-V cells as WT which is OK. But the wild-type of MCF7-V cells is MCF7 which has been defined in line 115. Even though you did not define wild-type as WT, it is still confusing and misleading. I’d suggest the authors use some other descriptions, for example, control. Or if the authors prefer to use WT, please homogeneous through the whole manuscript, for example in Figure 3J the authors use WT, while in Figure 3K, 3L, and 3M the authors use wt.

Round 2

Reviewer 1 Report

The manuscript has substantially improved after revisions suggested by the reviewers.

Author Response

Thank you very much for your positive feedback

Reviewer 4 Report

PLEASE BE AWARE OF STATISTICS. This is not only a sentence in your method or figure legend.

  1. Page 2 line 44-46: Since it is already 2022, I’d like to suggest the authors update the data to 2021 if you can.
  2. Throughout the whole manuscript, please update the CAT# to the reagents you used in this study. Clear sources of reagents might make your study more reproductive by other researchers.
  3. Line 291: “A two-tailed PAIRED t-test” the authors please point out where/which figure you use this.
  4. Page 10 line 379: The p italic for the ****p ≤0.0001, p.
  5. Figure 2: The authors analyzed Figure 2C, 2G, and 2I with two-way ANOVA, that’s no problem. Now let’s check Figure 3 (first round review comments: major 4), the authors please clarify, did you use two-way ANOVA to analyze Figure 3D, 3E, 3F, 3G, 3H, 3I, 3L, and 3M? How? What’re the multiple comparisons, please? What’re the row effect and column effect please if the two-way ANOVA was used to analyze the data as indicated above?
  6. Figure 3 : (First round review comments: major 5) The authors indicated that n≥3 for panels D-I in Figure 3. I’d like to confirm with you in detail, Fig 3D WT vehicle, Fig 3E WT vehicle, Fig 3F WT in both vehicle, ICI and shSPRED2_1 in ICI, Fig 3G WT in both vehicle, ICI and shEV in ICI, Fig 3H WT vehicle and shScrambled in E2, and Fig 3I WT vehicle, these groups seem only with one sample in each group, do you mean these groups with no less than 3 samples in each and all the samples with same values, that’s why they overlapped as one? 
